# Factors Differentiating Rural and Urban Population in Determining Anxiety and Depression in Patients with Chronic Cardiovascular Disease: A Pilot Study

**DOI:** 10.3390/ijerph18063231

**Published:** 2021-03-20

**Authors:** Elżbieta Szlenk-Czyczerska, Marika Guzek, Dorota Emilia Bielska, Anna Ławnik, Piotr Polański, Donata Kurpas

**Affiliations:** 1Department of Health Sciences, University of Opole, 68 Katowicka Street, 45-060 Opole, Poland; 2Medical and Diagnostic Centre (MDC), 9 Niklowa Street, 08-110 Siedlce, Poland; marika.guzek@centrum.med.pl; 3Department of Family Medicine, Medical University of Białystok, 1 J. Kilińskiego Street, 15-089 Białystok, Poland; d.bielska1@wp.pl; 4Faculty of Health Sciences, Pope John Paul II State School of Higher Education, 95/97 Sidorska Street, 21-500 Biala Podlaska, Poland; lawnikania@gmail.com; 5Family Physician’s Practice, Non-Public Healthcare Center, 4 Nad Potokiem Street, 58-350 Mieroszow, Poland; p.polanski@wp.pl; 6Department of Family Medicine, Wrocław Medical University, 1 Syrokomli Street, 51-141 Wrocław, Poland; dkurpas@hotmail.com

**Keywords:** chronic cardiovascular diseases, anxiety and depression, rural and urban population

## Abstract

The aim of this cross-sectional study was to analyze selected variables differentiating rural from urban populations, as well as identify potentially increased levels of depression and anxiety in patients with chronic cardiovascular disease. The study was carried out in 193 patients. The study used the Camberwell Assessment of Need Short Appraisal Schedule (CANSAS), the Health Behavior Inventory Questionnaire (HBI), the WHOQOL-BREF Quality of Life Questionnaire, and the Hospital Anxiety and Depression Scale–Modified Version (HADS-M). Spearman’s rank correlation coefficient test and logistic regression were used for analyses. In rural patients, we observed a relationship between anxiety and age (1/OR = 1.04; 95% CI: 0.91–0.99), the assessment of satisfied needs (1/OR = 293.86; 95% CI: 0.00001–0.56), and quality of life (QoL) in physical (OR = 1.56; 95% CI: 1.11–2.33), social (1/OR = 1.53; 95% CI: 0.04–0.94), and environmental domains (OR = 1.67; 95% CI: 1.06–3.00), as well as between depression and QoL in physical (1/OR = 1.39; 95% CI: 0.50–0.97) and psychological (OR = 1.37; 95% CI: 1.01–1.93) domains. In city patients, we observed a relationship between the drug and Qol in the physical (1/OR = 1.25; 95% CI: 0.62–0.98) and psychological (OR = 1.49; 95% CI: 1.13) domains. Younger patients living in a rural area with a lower assessment of met needs, a higher level of QoL in physical and environmental domains, and a lower social domain, as well as patients living in a city with a lower QoL in the physical domain and a higher psychological domain, have a greater chance of developing anxiety and depressive disorders.

## 1. Introduction

Cardiovascular diseases (CVDs) decrease mobility and reduce work and social activity, and thus are a cause of social isolation. This health condition may also result in chronic anxiety, depressed mood, and, consequentially, development and aggravation of depressive and anxiety syndrome [1,2,3]. 

It is estimated that anxiety disorders are found in 28–44% of young people and 14–24% of older people with CVDs [4]. Conversely, patients with anxiety disorders have also been noted to have a higher chance of developing coronary artery disease [5]. The risk of death caused by ischemic heart disease is doubled if patients develop panic disorder as well [6]. A high level of anxiety has also been proven to be a significant predictor of recurrent heart incidents [7,8]. High emotional states and levels of anxiety might directly lead to the development of sudden cardiological incidents as well [9]. 

The incidence of depression in CVD patients is twice as high as in the general population, and results in a worse medical prognosis for these patients [10]. One out of five patients with ischemic heart disease or heart failure suffers from depression, three times as high as in the general population. Depression affects an even higher percentage of post-stroke patients (one out of three) [2]. Depressed mood is found in 66.6% of hospitalized patients who have experienced a heart infarct, while depression is diagnosed in 15%. Patients with chronic heart failure are even more prone to depression (10–40% of cases) [11,12]. Depression in patients suffering from coronary heart disease is the strongest prognostic factor of death (depressive patients have a doubled risk of death over patients without depression) [13]. This state is also related to the aggravation of functional disorders, lower therapeutic compliance, and reduced participation in cardiological rehabilitation [14]. It was shown that patients with depression develop CVD in the subsequent six years two or three times more often [15]. In Poland, depression is diagnosed in 5–16% of the population, with symptom severity increasing with age [16]. The economic and health-related indicators of CVDs and depression result in higher treatment costs, higher demand for health services, and decreased efficiency [11].

Although diagnosis of depression is steadily increasing (it is currently the third most common cause of all GP appointments), a great number of cases remain unnoticed or are treated late [17]. Research shows that depression is found less frequently in depressive CVD patients than in patients with similar symptoms, but not suffering from cardiovascular disease [18]. 

As mental condition has a huge impact on treatment and prognosis, the need for better identification strategies is increasingly recognized. The disorders ought to be diagnosed earlier, and appropriate treatment should be given [19,20,21]. Anxious and depressive patients without social support whose symptoms of diseases do not progress might develop an unfavorable attitude towards the disease, feeling helplessness and anxiety. The symptoms of anxiety and depression decrease the motivation for changing lifestyle, and tend towards social isolation and even mental escape from crucial issues [22]. Taking into consideration the role of primary healthcare, it is important to examine the epidemiology of anxiety and depressive disorders thoroughly, as well as the risk factors for their occurrence in CVD patients in relation to Poland’s primary health services. To our knowledge, this is the first study conducted in Poland to evaluate the determinants of anxiety and depression and to differentiate rural and urban populations of patients with CVDs under the care of primary care family nurses.

Considering the above, the purpose of this study was to determine whether there are variables affecting anxiety and depressive symptoms in patients with cardiovascular disease that differentiate the rural and urban population and potentially determine the increase in the effectiveness of primary care. The results of this study may be a source of information in the process of creating systems to identify risk groups of lower care effectiveness among patients with chronic cardiovascular diseases, as well as a resource for clinical information systems and support in making decisions as part of designing personalized care models and a system for identifying people with chronic cardiovascular diseases residing rural and urban areas requiring psychological support.

## 2. Materials and Methods

### 2.1. Study Design

This study is a cross-sectional observational study. This study is part of a broader study to identify indicators that determine the effectiveness of home care for patients with chronic CVDs, and to identify variables that determine effective support systems for their home caregivers. The study involved 350 patients with CVDs. In order to define indicators specific to home care, 193 patients remained in the home under the care of primary care family nurses, while 157 patients went out to see their GP for follow-up visits. The study also included caregivers of patients under home care of primary care family nurses. The study involved 161 caregivers. This article presents a partial analysis of the results of this study concerning variables affecting anxiety and depressive symptoms in patients with cardiovascular disease under the home care of a primary care family nurse. 

### 2.2. Sample

The study was conducted in Polish CVD patients. These patients received home care from a family nurse working in basic healthcare in the Opolskie (Opole), Dolnośląskie (Wrocław, Mieroszów), Mazowieckie (Siedlce), Lubelskie (Biała Podlaska), and Podlaskie (Białystok) provinces. Eight primary care institutions took part in the study. Patients were encouraged to take part in the study by a family nurse during planned home visits. Respondents completed questionnaires in their home. Patients were provided with one set of questionnaires each, and nurses completed an additional questionnaire concerning the patient (i.e., paired questionnaires with respect to the same patient). Data was collected from March 2016 to January 2017.

In our study, we used a non-probabilistic sampling method (purposive sampling). Two hundred CVD patients under the home care of family nurse practitioners were invited to participate in the survey. The final sample of participants was determined on the basis of their temporal availability. Finally, 193 people took part in the survey. The criteria for inclusion in the study were as follows: 18 years, confirmation of chronic cardiovascular disease at least 12 months prior to the study, and staying under the home care of a family nurse. Chronic CVD was defined and ascertained based on the medical history of primary healthcare. The exclusion criteria (disqualification as determined by the family nurse) were cognitive disorders and other severe mental illness and/or other difficulties preventing active participation in the study.

Participation in the study was voluntary and anonymous. All participants were informed about the study aims, methods, and the ability to withdraw participation at any stage of the examination. 

### 2.3. Ethical Aspects 

The study was approved by the Bioethical Commission at the Medical University in Wroclaw (No KB-86/2016).

### 2.4. Variables and Data Collection 

To evaluate the study patients’ quality of life, a short Polish version of the World Health Organization Quality of Life (WHOQOL) questionnaire was used. It measures the QoL in four domains: physical, psychological, social relations, and environment. The tool also includes some separately analyzed questions relating to the individual perception of quality of life (question 1) and the perception of each patient’s health condition (question 2). All responses used the five-degree Likert scale. The reliability of the Polish version of WHOQOL-BREF was checked with the use of Cronbach’s coefficient α, which was 0.81 for the physical domain, 0.78 for the psychological domain, 0.69 for the social relations domain, and 0.77 for the environment domain. The internal test coherence for the whole questionnaire was 0.90 [23,24].

To assess the level of health behavior of patients with CVDs, we used the Health-Related Behavior Inventory by Juczynski (HBI), which consists of 24 statements measuring four pro-health behavior categories, i.e., proper eating habits, preventive behaviors, proper mental attitudes, and health practices. The respondents determine the frequency of health behaviors and proper activity based on the scale in which 1 corresponds to hardly ever, 2 = rarely/seldom, 3 = from time to time, 4 = often, and 5 = usually/almost always. The values marked by examinees are then summed up to calculate an overall measure of health activity intensity, with a value ranging from 24 to 120 points. The higher the value, the higher the intensity of pro-health behaviors. Additionally, the intensity of all four categories is analyzed. The indicator equals the average number of points in each category, and is calculated by summing up all of the points and dividing by 6. The reliability of the test for all four subscales ranges from 0.6 to 0.65 for a total of 0.85 (Cronbach’s α) [25].

The levels of satisfied and unsatisfied needs were assessed with the modified version of the Camberwell Assessment of Need (CAN). The Camberwell Modified Short Needs Assessment questionnaire was developed by focus groups and competent judges, and was applied to evaluate the needs of ED and general practice patients. The modification of CAN is focused on 22 problem fields. In this research, the Camberwell Needs Index was calculated. The calculations consisted of the determination of the number (N) of met (1) and unmet (0) needs of the patient on the basis of 24 questions identifying 22 needs. Consecutively, within the number (N) of needs indicated by the studied person, the number (M) of met needs (1) was established. The M/N formula was used to calculate the Camberwell Index. In addition, the above method also makes it possible to calculate the Camberwell Index for unmet needs according to the formula 1–M/N (not used in this analysis). The consistency of Cronbach’s alpha for the modified version of the CAN questionnaire was 0.82 [26].

To assess anxiety and depression, a modified version of the HADS questionnaire (Hospital Anxiety Depression Scale), or HADS-M, was applied. The tool uses seven positions to measure anxiety, seven to evaluate depression level, and two to measure nervousness and aggression. The questionnaire is helpful to evaluate anxiety, depression, and aggression both in hospitalized and ambulatory patients. It contains 16 test questions assigned to points ranging from 0 to 3. The result of the test is obtained from summing up all of the points in each category. The maximum result for anxiety and depression is 21, while it is 6 for aggression. The anxiety and depression subscales are interpreted as follows: 0–7 points represent standard behavior, 8–10 represent border values indicating mild anxiety, while 11–21 are regarded as pathological, and indicate an anxiety syndrome/disorder. The validation examinations of the primary and modified version of the HADS scale prove its reliability and validity. The Spearman’s correlation coefficient between the test positions and the overall result of a given subscale was statistically significant (at least *p* < 0.01), and ranged from 0.41 to 0.76. The validity of the test was achieved by comparing the HADS scale results to the interview-based evaluation. The correlation coefficient for the anxiety subscale was found to be 0.54 and, for the depression subscale, it was found to be 0.79 [27,28].

To evaluate essential socio-demographic features of CVD patients, as well as the number and type of health services provided to them, the author’s interview questionnaire was used to obtain information about sex, age, marital status, level of education, financial condition, place of residence, the number of hospitalizations within the prior three years, the number of appointments at GPs in primary health institutions within the prior 12 months, the number of visits to a cardiology clinic within the prior 12 months, the number of family nurse interventions within the prior 12 months, and the number of GPs/nurse home visits in the prior 12 months.

### 2.5. Data Analysis

The results of the study were subject to statistical analysis with the use of the R statistical package (version 3.4.0, Publisher, City, Postal code, Country).

For the quantitative variables, the arithmetic mean, standard deviation, first quartile (Q.25%), median (Q.50%), third quartile (Q.75%), minimum, and maximum were calculated. For nominal variables, frequency (i.e., percent) was determined. The Shapiro–Wilk test showed that only a few variables had a standard normal distribution, namely WHOQOL-BREF in physical and psychological domains, as well as the environment. The distributions of the other variables were totally different from the standard normal one. Therefore, the Chi-square test, Fisher’s test, Wilcoxon’s test, and Spearman’s rank correlation coefficient were used for further analysis. The test probability at the level of *p* ≤ 0.05 was considered significant. The relationship between the number of visits, advice, and interventions over the last 12 months with the HADS-M was analyzed using Spearman’s rank correlation coefficient, which does not require a normal distribution of the variables. The null hypothesis (H0) was tested, wherein the Spearman’s rank correlation coefficient equaled 0. The alternative hypothesis was that the correlation coefficient differs from 0. The null hypothesis (H0) was rejected if the *p*-value was <0.05 (α = 0.05). 

Table 1 shows the number of GP appointments, visits in the cardiological clinic, and the number of interventions by the family nurse practitioner in the past 12 months by place of residence, including city and village (see Table 1).

To examine the relationship between anxiety or depressive symptoms and selected variables, a logistic regression analysis was used. Two dependent variables were analyzed. The individual analysis of each double-value explained variable was conducted as shown below:HADS-M Anxiety: 0—lack of abnormalities (0–10), 1—abnormality present (11–21)HADS-M Depression: 0—lack of abnormalities (0–10), 1—abnormality present (11–21).

The describing variables for the logistic regression analysis models were selected from the group of the following 29 variables (see Table 2).

Then, a separate logistic regression analysis was carried out for both dependent variables examining all possible models derived from, at most, six explanatory variables and the explained one. For further analysis, only the models with specific characteristics were chosen. All variables in the model had to be statistically significant, and the set of explanatory variables had to be the biggest possible, while the number of models the smallest (the study does not present the results of the logistic regression or the odds ratio in the model of logistic regression in the group of urban residents with the variable HADS-M depression test because of a very small sample). Using the models selected, the odds ratio for the events examined were calculated, and conclusions were formulated on their basis.

This way of proceeding did not require procedures to select variables for the model. In the present study, we did not rule out autocorrelation between depression and anxiety, because these variables correlated poorly with each other. In the total group, the Spearman rank correlation coefficient was *r* = 0.22 (*n* = 179, *p* = 0.003), in the urban group *r* = 0.21 (*n* = 106, *p* = 0.034), and in the rural group *r* = 0.26 (*n* = 73, *p* = 0.028). Therefore, we considered it appropriate to examine these variables separately.

Only respondents with no missing data in all variables used in the given calculations were selected for statistical calculations.

## 3. Results

### 3.1. Sociodemographic Data of Patients with CVDs

Most of the CVD patients were women. The examinees were mostly women (71.7%; *n* = 81) living in the city vs. a village (65.8%; *n* = 50). The median age was 76 years for city residents (min–max: 17–101 years) and 72 (min–max: 18–94 years) for village residents (see Table 3).

### 3.2. The Evaluation of Anxiety and Depression

The analysis of anxiety and depression occurrence in the group of patients, calculated into a standard ten, is presented in Table 3. A pathological disorder, such as anxiety syndrome, was found in 61.1% (*n* = 72) of patients living in cities and in 68.9% (*n* = 51) of rural residents. Low intensity (mild) anxious behaviors were observed in 32.1% (*n* = 35) city vs. 28.4% (*n* = 21) village residents. A lack of abnormalities was confirmed in 1.8% (*n* = 2) city vs. 2.7% (*n* = 2) village residents. The analysis of the HADS-M depression variable found depression in 68.8% (*n* = 75) of city residents vs. 68% (51) of village residents. Mild depressive behaviors were found in 26.6% (*n* = 29) city vs. 28% (*n* = 21) village residents. A lack of depression was confirmed in 4.6% (*n* = 5) city vs. 4% (*n* = 3) village residents. A lack of significant differences between city and village residents was observed at the level of significance equal to 0.05 (see Table 4).

### 3.3. Correlations of the Number of Visits and Interventions in the Prior 12 Months with the HADS-M Scale

Statistically significant differences between the correlation coefficients of city and village residents were found relating to anxiety and the number of visits in cardiological clinics (*p* = 0.005) and the number of interventions of a family nurse (*p* = 0.03). In village patients, the number of visits to cardiological units (*r* = 0.32, *p* = 0.005) and the number of nurse interventions (*r* = 0.25, *p* = 0.033) correlated positively with the HADS-M anxiety scale. This correlation was not found in city patients. The correlation between the number of visits to cardiological units and HADS-M anxiety was found to be insignificant (*r* = −0.1, *p* = 0.306), as well as between the number of nurse interventions and HADS-M anxiety (*r* = −0.06, *p* = 0.516) (see Table 5).

### 3.4. Results of Logistic Regression

The logistic regression analysis in the group of CVD patients living in urban areas led to the selection of the models, which allowed for the odds ratio calculation. Table 6 presents the results of the odds ratio in a logistic regression model for the risk of anxiety occurrence in chronic CVD patients living in cities.

It was found that, in patients who differed at the level of QoL in the physical domain of the WHOQOL-BREF questionnaire by one degree, those with a lower score had a 1.25 times higher chance of anxiety occurrence than those with a higher result. In patients who differed in this score by 14.85, those with a lower one had a 30.93 times higher chance of developing anxiety.

In patients who differed in the level of QoL in the psychological domain of the WHOQOL-BREF questionnaire by one degree, those with a higher score had a 1.49 times higher chance of anxiety occurrence than those with a lower one. In patients who differed in this score by 14.66, those with a higher result had a 48.75 times higher chance of anxiety abnormalities.

It was found that, in patients who differed in the level of QoL in the intensity of health practices of the WHOQOL-BREF questionnaire by one degree, those with a lower score had a 1.79 times higher chance of anxiety occurrence than those with a higher one. In patients who differed in this score by 2.83, those with a lower one had a 5.21 times higher chance of developing anxiety.

It was also discovered that, for patients who differed in the assessment of depression in the HADS-M scale by one degree, those with a higher score had a 4.07 times higher chance of anxiety occurrence than those with a lower one (see Table 6).

The logistic regression analysis in the group of CVD patients living in rural areas led to the selection of models, which allowed for odds ratio calculations (see Table 7). 

It was found that, among patients who differed in age by one year, younger patients had a 1.04 times greater chance of developing an anxiety disorder than older patients, while in patients who differed in age by 76 years, the younger patients had a 21.2 times greater chance.

In patients who differed in the Camberwell evaluation of needs by 0.83, those with a lower score had a 113 times higher chance of anxiety occurrence than those with a higher assessment. In patients who differed in the assessment by one, those with a lower assessment had a 293.86 times higher chance of such abnormalities.

In patients who differed in the level of QoL in the physical domain of the WHOQOL-BREF questionnaire by one degree, those with a higher score had a 1.49 times higher chance of developing anxiety than those with a lower one. In patients who differed in this assessment by 14.28, those with a higher assessment had a 585 times higher chance of anxiety disorders.

In patients who differed in the level of QoL in the social relations domain of the WHOQOL-BREF questionnaire by one degree, those with a lower score had a 1.53 times higher chance of anxiety occurrence than those with a higher one. In patients who differed in this assessment by 14.66, those with a lower one had a 522 times higher chance of anxious behaviors.

It was also discovered that, in patients who differed in the level of QoL in the environment domain of the WHOQOL-BREF questionnaire by one degree, those with a higher score had a 1.67 times higher chance of anxiety occurrence than those with a lower one. In patients who differed in this assessment by 11.92, those with a higher one had a 473 times higher chance of anxiety disorders.

It was confirmed that, in patients who differed in the assessment of depression by one on the HADS-M scale, those with a higher score had a 5.85 times higher chance of anxiety disorders than those with a lower score.

In patients who differed in the number of family nurse visits in the last 12 months by one, it was found that those with a lower number of visits had a 1.11 times higher chance of anxiety development than those with a greater number of visits. However, in patients who differed in the number of visits by 28, those with a smaller number had a 51.39 times higher chance of such behaviors.

For patients who differed in the level of QoL in the psychological domain of the WHOQOL-BREF questionnaire by one degree, those with a higher score had a 1.32 times higher chance of anxiety occurrence than those with a lower score. However, in patients who differed in this assessment by 14, those with a higher score had a 51.39 times higher chance of developing anxiety (see Table 7).

The analysis of logistic regression in chronically ill CVD patients living in rural areas led to the identification of models that permitted the calculation of the odds ratio. Table 8 presents the results of the odds ratio in the model of logistic regression for the risk of depression in chronically ill CVD patients living in rural areas.

In patients who differed in the level of QoL in the physical domain of the WHOQOL-BREF questionnaire by one degree, those with a lower score had a 1.39 times higher chance of depression than those with a higher score. In patients who differed in this assessment by 14.28, those with a lower score had a 118.39 times higher chance of developing such conditions.

In patients who differed in the level of QoL in the psychological domain of the WHOQOL-BREF questionnaire by one degree, those with a higher score had a 1.37 times higher chance of depressive behaviors than those with a lower score. For patients who differed in this assessment by 14, those with a higher score had a 89.07 times higher chance of such abnormalities.

It was also found that, in patients who differed in the assessment of anxiety on the HADS-M scale by one, those with a higher score had a 4.6 times higher chance of depression than those with a lower score (see Table 8). 

## 4. Discussion

In our study, we analyzed selected variables that may have an impact on improving the quality of health services provided to patients with CVDs living in urban and rural areas and with elevated levels of anxiety and depression. Previous research has found that CVDs disproportionately affect females [29]. The results prove the conclusion above, as women were the majority of respondents both in cities (71.7%; *n* = 81) and villages (65.8%; *n* = 50). The study does not include an analysis of anxiety and depression occurrence in relation to sex, however, the results support the conclusion that these conditions affect women more than men, regardless of place of residence. The results achieved are compatible with other studies conducted in Poland and in other countries [16,30]. It appears that being female is a high risk factor for depression and/or anxiety in the CVD patient population, which requires close monitoring and further research regarding women vs. men. 

Researchers have found that the emotional state of patients, together with other standard factors, constitutes a crucial prognostic element for the course of a disease. The circulatory system is sensitive to psychic stimuli, and its functioning affects the emotional state and central nervous system. The reverse relationship also exists; emotional disturbances related to the diagnosis of a CVD might negatively influence the general quality of a patient’s life [22,31]. Anxiety and depression are common in CVD patients [2,4,32]. The results of this study correlate with these findings. The study found anxiety disorders in 66.1% (*n* = 72) of patients living in cities and 68.9% (*n* = 51) of those living in villages. Depression was observed in 68.8% (*n* = 75) of urban respondents and 68% (*n* = 51) of rural respondents. No statistically significant differences between city and village residents were found.

Anxiety has a negative influence on the prognosis of diagnosed CVD patients relating to the increased risk of a series of cardiovascular incidents, such as stroke or death caused by cardiovascular failure [7,8,9]. In analyzing the results of the study, we noted that the number of family nurse visits and visits in cardiological clinics among village residents correlated positively with the HADS-M anxiety scale. No such correlation was observed in patients living in the city. Proper emotional support is one of the most essential elements in preparing a patient for information about their condition, as it decreases anxiety and stress and consequently increases quality of life [33]. Among the various care situations CVD patients have, systematic self-control and check-ups in medical clinics or carried out at patients’ homes seem to be the most important. The form of education depends on the opportunities of a medical institution. However, the more interactive and diverse it is, the higher its effectiveness. When patients are fully aware of their condition and possess proper knowledge about it, anxiety decreases [34]. It is worth highlighting that living in rural areas offers fewer possibilities and limits access to health services, support groups, and educational programs, thus lowering the safety level in the patients [35]. The results gathered in the study indicate the necessity for more attention, while creating programs aimed at early prevention of anxiety and education among village patients who visit cardiological clinics more often and receive more nurse interventions at home.

Circulatory diseases are very common, and thus constitute a specific area of research attention on the quality of life. Determining physical, psychological, and social consequences of CVDs on the life of an individual should remain a core interest. Numerous studies have confirmed that QoL assessment is as important in this group of patients as physical, laboratory, and clinical tests [36]. Living in a rural environment is believed to be a strong identification of the quality of life connected with health [37]. It was proved in this study that the risk of anxiety among CVD patients living in villages is related to a higher score of QoL in the physical, psychological, and environmental domains, and a lower score in the social domain, while a higher risk of depression is indicated by a lower score in the physical domain and a higher score in the psychological domain of QoL. Among city patients, the higher risk of anxiety was determined by a lower score in the physical domain and a higher score in the psychological domain of QoL. It is notable that CVD patients evaluating their mental state higher are more prone to anxiety disorders and depression, and need psychological support.

Family nurse visits are essential in the course of care of CVD patients. Van Spall et al., while evaluating the effectiveness of home visits in patients with heart failure and cared for by nurses, found that visits were the most powerful strategy to decrease mortality and readmissions after hospitalization caused by heart failure [38]. Thereby, they confirmed earlier findings made by Felter et al. [39]. This study showed unequivocally that more home visits are associated with a lower risk of anxiety occurrence in CVD patients.

Healthcare behaviors are viewed as the main element in CVD prevention. The World Health Organization [40], the American Heart Association (AHA) [41], and European guidelines related to the prevention of CVD in clinical practice [42] underline the value of healthcare behaviors in preventing and decreasing CVD morbidity. The study showed that a lower intensity of pro-health behaviors might foster the risk of anxiety disorders as in patients living in cities who differed in the intensity of the behaviors by one degree. Those with a lower score have a 1.79 higher chance of anxiety occurrence than those with a higher score.

It was also confirmed that the presence of anxiety and depression among patients, especially elderly ones, is more common in rural than urban areas [43]. In our study, it was observed that younger age and village residence determines a higher risk of anxiety disorders. Our results suggest the need for further research in this area.

The relationship between the level of met needs and the risk of anxious behaviors might also be interesting. The analysis of healthcare systems concerning primary health care over the chronically ill emphasize the issue of health needs. This is believed to be an outcome of the level of a clinical condition and factors deriving from it, such as the quality of life, healthcare behaviors, and the evaluation of medical services. It is assumed that the recognition of a need is equal to the identification of a problem, and allows for proper intervention [26,44]. In the process of shaping primary health systems, determining the individual biopsychosocial needs of patients is becoming more and more essential [44]. We found here that when the level of needs met in patients living in rural areas decreases, the risk of the occurrence of anxiety increases.

Examining the relationship between anxiety and depression requires particular attention in the discussion about select variables affecting the improvement of primary healthcare of CVD patients who experience these abnormalities. We found that a higher score in the HADS-M depression scale is related to a higher risk of anxiety occurrence in CVD patients, regardless of the place of residence. What is more, a higher score in the HADS-M anxiety scale increases the risk of depression in rural patients. The characteristics presented here might constitute the basis for deeper research into the concepts, and shows a huge need for professional support. Comorbidities negatively impact the quality of life and disability in patients with chronic diseases [45,46]. The comprehensive management of patients with CVD should include the treatment of comorbidities that have been associated with a worse quality of life, with special emphasis on anxiety and depression disorders. An adequate control of comorbidities may have a positive impact on the quality of life of patients. Patients who fit the characteristics ought to be targeted with medical and social programs that ensure their stable condition and improve the quality of their lives. 

This study may be limited by a small sample size. This could significantly limit the possibility of the generalization of the results to the whole population of CVD patients in Poland. The findings discovered in the study, however, remain valuable, and might be used in the course of interventions supporting the development of a systemic model of home care over chronically ill patients. Research using a greater number of CVD patients and healthcare institutions in urban and rural areas is encouraged.

## 5. Conclusions

In our study, we observed that variables affecting anxiety in patients living in rural areas include: younger age, a higher number of cardiology outpatient clinic visits, a higher number of family nurse interventions, a lower number of home visits, a lower needs assessment, a higher level of QoL in the physical, environmental, and psychological domains, and lower QoL in the social domain. Depressive disorders may be more common in patients living in rural areas, with lower QoL in the physical domain and higher QoL in the psychological domain. Patients living in urban areas with lower QoL in the physical domain and higher QoL in the psychological domain, lower health behavior scores in the health practices category, and with depressive disorders have a higher chance of suffering from depressive disorders.

## Figures and Tables

**Table 1 ijerph-18-03231-t001:** Distribution of visits, advice, and interventions over the last 12 months.

Variables	Values	ALL	City	Village
		(*n* = 193)	(*n* = 113)	(*n* = 76)
GP appointments	Min.	0	0	0
	1st Qu.	2	2	3
	Median	5	5	4.5
	Mean	6.4	6.9	6.2
	3rd Qu.	10	10	9.3
	Max.	60	60	20
	NA’s	-	4	4
Visits in a cardiological clinic	Min.	0	0	0
	1st Qu.	0	0	0
	Median	1	1	1
	Mean	1.8	1.7	1.9
	3rd Qu.	2	2	2
	Max.	24	15	24
	NA’s	-	4	4
Family nurse interventions	Min.	0	0	0
	1st Qu.	0	0	0
	Median	4	4	9
	Mean	16.4	19.9	12.0
	3rd Qu.	12	12	15
	Max.	198	198	48
	NA’s	-	4	4

Legend: GP—general practitioner, Max.—maximum, Qu.—Quartile, NA’s—Missing data.

**Table 2 ijerph-18-03231-t002:** Descriptive variables for logistic regression analysis models.

No.	Variables	Coding
X1	Weight (kg)	
X2	Growth (cm)	
X3	Does the patient receive social benefits from a social assistance center?	1—Yes 2—No
X4	Number of home visits in the last 12 months	
X5	Number of telephone consultations in the last 12 months	
X6	Number of visits to PHC clinics during the last 12 months	
X7	Number of family nurse interventions in the last 12 months	
X8	Sex	1—Woman 2—Man
X9	Age (in years)	
X10	Education	1—Primary 2—Vocational 3—Secondary without Matura Exam 4—Secondary with Matura Exam 5—Post-secondary 6—Higher (BA/MA)
X11	Financial situation	1—Very good (above PLN 3001 per person in the family) 2—Good (from PLN 2001–3000 per person in family) 3—Average (from PLN 1001–2000 per person in family) 4—Bad (from PLN 501–1000 per person in family)
X12	Number of visits to the GP in the last 12 months	
X13	Number of visits to the cardiology outpatient clinic during the last 12 months	
X14	Number of family nurse interventions in the last 12 months	
X15	Camberwell	
X16	WHOQOL-BREF question 1	
X17	WHOQOL-BREF question 2	
X18	WHOQOL-BREF physical domain	
X19	WHOQOL-BREF psychological domain	
X20	WHOQOL-BREF domain social relations	
X21	WHOQOL-BREF domain environment	
X22	HBI sum	
X23	HBI proper eating habits	
X24	HBI preventive behaviors	
X25	HBI proper mental attitudes	
X26	HBI health practices	
X27	HADS-M Aggression	
X28	HADS-M Anxiety or HADS-M Depression	0—Lack of abnormalities (0–10) 1—Abnormality present (11–21)
X29	Is he/she in a relationship?	1—No 2—Yes

Legend: No.—Variable number, WHOQOL-BREF—the WHOQOL-BREF Quality of Life Questionnaire, HBI—the Health Behavior Inventory Questionnaire, HADS-M—the Hospital Anxiety Depression Scale–Modified Version.

**Table 3 ijerph-18-03231-t003:** Socioeconomic situation in the respondent group.

Variable	Place of Residence	*n*	Q.25%		Me	Q.75%		Min	Max	Wilcoxon Test *p*
Age (in years)	City	113	62.00		76.00	84.00		17.00	101.00	0.202
	Village	75	62.50		72.00	81.50		18.00	94.00	
**Variable**	**Categories**		**City**			**Village**			**Fisher Test—*p***	
			***n***	**%**		***n***	**%**			
Sex	Women		81	71.7		50	65.8		0.423	
	Men		32	28.3		26	34.2			
	Total		113	100		76	100			

Legend: *n*—group quantity, %—percentage; M—mean; Q.25%—first quartile; Me—median; Q.75%—third quartile; Min.—minimum; Max.—maximum; *p*—calculated level of significance for standard Fisher’s test. The figures in column *n* do not sum up to 193 due to gaps in the questionnaires completed by the respondents.

**Table 4 ijerph-18-03231-t004:** The level of anxiety and depression by place of residence.

Variable	Place of Residence	*n*	Q.25%	Me	Q.75%	Min	Max	Wilcoxon Test *p*
HADS-M Anxiety	City	109	2.29	10.00	12.00	13.00	6.00	0.491
	Village	74	2.26	10.00	12.00	13.00	6.00	
HADS-M Depression	City	109	10.00	12.00	13.00	5.00	17.00	0.64
	Village	75	10.00	11.00	12.00	7.00	15.00	
**Variable**	**Categories**		**City**		**Village**		**Fisher Test—*p***	
			***n***	**%**	***n***	**%**		
HADS-M Anxiety	Lack of Abnormality (0–7)		2	1.8	2	2.7	0.78	
	Border Conditions (8–10)		35	32.1	21	28.4		
	Abnormality confirmed (11–21)		72	66.1	51	68.9		
	Total		109	100	74	100		
HADS-M Depression	Lack of abnormality (0–7)		5	4.6	3	4	0.962	
	Border Conditions (8–10)		29	26.6	21	28		
	Abnormality confirmed (11–21)		75	68.8	51	68		
	Total		109	100	75	100		

Legend: *n*—group quantity, %—percentage; Q.25%—first quartile; Me—median; Q.75%—third quartile; Min.—minimum; Max.—maximum; *p*—calculated level of significance for the standard Fisher test. The figures in column *n* do not sum up to 193 due to gaps in the questionnaires completed by the respondents.

**Table 5 ijerph-18-03231-t005:** The correlation between the number of visits, appointments, and interventions in the prior 12 months with the HADS-M scale.

Variable	Place of Residence	GP Appointments						Visits in Cardiological Clinic						Family Nurse Interventions					
HADS-M		*r*	*r* = 0 *p*	*	*n*	*r* = *r p*	*	*r*	*r* = 0 *p*	*	*n*	*r* = *r p*	*	*r*	*r* = 0 *p*	*	*n*	*r* = *r*	*
Anxiety	City	−0.11	0.244		109	0.439		−0.1	0.306		109	0.005	*	−0.06	0.516		109	0.03	*
	Village	−0.23	0.051		74			0.32	0.005	*	74			0.25	0.033	*	74		
Depression	City	−0.21	0.031	*	109	0.694		−0.03	0.739		109	0.37		−0.05	0.628		109	0.35	
	Village	−0.15	0.204		75			0.1	0.374		75			0.09	0.427		75		

Legend: (*r*)—Spearman’s rank correlation coefficient (0 means *r* ≤ 0.01), (*r* = 0 *p*)—calculated significance level for test verifying the null hypothesis that the correlation coefficient *r* equals 0; * appears in column (*) if *p* ≤ 0.05, then the null hypothesis is rejected (0 means *p* < 0.001); (*r* = r *p*)—calculated significance level for test verifying the null hypothesis that the two correlation coefficients are equal; * appears in column (*) if *p* ≤ 0.05, then null hypothesis is rejected (0 means *p* < 0.001). The figures in column *n* do not sum up to 193 due to gaps in the questionnaires completed by the respondents.

**Table 6 ijerph-18-03231-t006:** The results of the logistic regression analysis and odds ratio of the logistic regression model in the group of city residents. Explained variable: HADS-M anxiety (0—lack of abnormalities, 1—abnormality confirmed).

Explanatory Variables		b_i_	SE_i_	z_i_	*p_i_* = Pr (>|z_i_|)			
Models with four explanatory variables								
**Model 1 (*n* = 100)**								
Chi^2^ = 25.57, df = 4, *p* < 0.001, pseudo R^2^ = 0.2								
No.	Intercept	-	-	-	-			
X18	WHOQOL-BREF physical domain	−0.231	0.116	−1.991	0.046			
X19	WHOQOL-BREF psychological domain	0.399	0.147	2.723	0.006			
X26	HBI—health practices	−0.583	0.225	−2.597	0.009			
X28	HADS-M depression	1.406	0.481	2.924	0.003			
**OR**		**Per unit**			**Per range**			
		**OR**	**95% CI**	**1/OR**	**OR**	**95% CI**	**1/OR**	**range**
**from Model 1**								
X18	WHOQOL-BREF physical domain	0.79	0.62–0.98	1.25	0.03	0.00–0.84	30.93	14.85
X19	WHOQOL-BREF psychological domain	1.49	1.13–2.01	0.67	48.75	6.17–29,800	0.00	14.66
X26	HBI—Health Practices	0.55	0.35–0.85	1.79	0.19	0.05–0.64	5.21	2.83
X28	HADS-M Depression	4.07	1.62–10.83	0.24	4.07	1.62–10.80	0.24	1.00

Legend: OR—odds ratio, CI—95% confidence interval for OR. Chi-squared—statistical hypothesis test of the chi-squared model adjustment; df—number of degrees of freedom; *p*—calculated level of test significance; pseudo R^2^ —value that evaluates explanatory variable anticipation according to the model; b_i_—coefficient estimation in regression model; SE_i_—standard error estimation for the b_i_ coefficient; z_i_—value of test statistics in a standard distribution; (*p_i_* = Pr (>|z_i_|)—calculated probability value *p_i_* for the double-sided critical area equal to z; *n*—group quantity.

**Table 7 ijerph-18-03231-t007:** The results of the logistic regression analysis and odds ratio of the logistic regression model in the group of village residents. Explained variable: HADS-M anxiety (0—lack of abnormalities, 1—abnormality confirmed).

Explanatory Variables		b_i_	SE_i_	z_i_	*p_i_* = Pr (>|z_i_|)			
Models with six explanatory variables								
**Model 1 (*n* = 72)**								
Chi^2^ = 40.05, df = 6, *p* < 0.001, pseudo R^2^ = 0.4								
No.	Intercept	-	-	-	-			
X9	Age (in years)	−0.040	0.020	−2.014	0.044			
X15	Camberwell	−5.683	2.741	−2.073	0.038			
X18	WHOQOL-BREF physical domain	0.446	0.185	2.405	0.016			
X20	WHOQOL-BREF social relations domain	−0.427	0.214	−1.997	0.046			
X21	WHOQOL-BREF environment domain	0.516	0.260	1.987	0.047			
X28	HADS-M depression	1.768	0.698	2.532	0.011			
Models with four explanatory variables								
**Model 2 (*n* = 73)**								
Chi2 = 24.39, df =4, *p* < 0.001, pseudo R2 = 0.24								
	Intercept	-	-	-	-			
X4	Number of home visits by a nurse during the last 12 months	−0.102	0.050	−2.059	0.040			
X15	Camberwell	−4.397	2.098	−2.096	0.036			
X19	WHOQOL-BREF psychological domain	0.281	0.124	2.276	0.023			
X28	HADS-M depression	1.201	0.559	2.148	0.032			
**OR**		**Per unit**			**Per range**			
		**OR**	**95% CI**	**1/OR**	**OR**	**95% CI**	**1/OR**	**range**
**from Model 1**								
X9	Age (in years)	0.96	0.919–0.99	1.04	0.05	0.002–0.73	21.2	76.00
X15	Camberwell	0.00	0.00001–0.56	293.86	0.01	0.00007–0.62	113	0.83
X18	WHOQOL-BREF physical domain	1.56	1.11–2.33	0.64	585	4.58–177,000	0.002	14.28
X20	WHOQOL-BREF social relations domain	0.65	0.40–0.94	1.53	0.002	0.000002–0.46	522	14.66
X21	WHOQOL-BREF environment domain	1.67	1.06–3.00	0.59	473	2.11–503,000	0.002	11.92
X28	HADS-M depression	5.85	1.58–25.66	0.17	5.85	1.58–25.60	0.17	1.00
**from Model 2**								
X4	Number of home visits by a nurse during the last 12 months	0.90	0.81–0.99	1.11	0.06	0.01–0.83	17.64	28
X19	WHOQOL-BREF psychological domain	1.32	1.05–1.71	0.75	51.39	2.03–1960.55	0.02	14

Legend: OR—odds ratio, CI—95% confidence interval for OR. Chi-squared—statistical hypothesis test of the chi squared model adjustment; df—number of degrees of freedom; *p*—calculated level of test significance; pseudo R^2^—value which evaluates explanatory variable anticipation according to the model; b_i_—coefficient estimation in the regression model; SE_i_—standard error estimation for the b_i_ coefficient; z_i_—value of test statistics in a standard distribution; (*p_i_* = Pr (>|z_i_|)—calculated probability value *p_i_* for the double-sided critical area equal to z; *n*—group quantity.

**Table 8 ijerph-18-03231-t008:** The odds ratio in the model of logistic regression and the odds ratio of the logistic regression model in the group of village residents. Explained variable: HADS-M anxiety (0—lack of abnormalities, 1—abnormality confirmed).

Explanatory Variables		b_i_	SE_i_	z_i_	*p_i_* = Pr (>|z_i_|)			
Models with three explanatory variables								
**Model 1 (*n* = 73)**								
Chi^2^ = 19.55, df = 3, *p* < 0.001, pseudo R^2^ = 0.19								
No.	Intercept	-	-	-	-			
X18	WHOQOL-BREF physical domain	−0.334	0.167	−2.003	0.045			
X19	WHOQOL-BREF psychological domain	0.321	0.162	1.984	0.047			
X28	HADS-M anxiety	1.528	0.609	2.507	0.012			
**OR**		**Per unit**			**Per range**			
		**OR**	**95% CI**	**1/OR**	**OR**	**95% CI**	**1/OR**	**range**
**from Model 1**								
X18	WHOQOL-BREF physical domain	0.71	0.50–0.97	1.39	0.01	0.00005–0.71	118.39	14.28
X19	WHOQOL-BREF psychological domain	1.37	1.01–1.93	0.72	89.07	1.31–10600	0.01	14.00
X28	HADS-M anxiety	4.60	1.45–16.28	0.21	4.60	1.45–16.2	0.21	1.00

Legend: OR—odds ratio, CI—95% confidence interval for OR. Chi-squared—statistical hypothesis test of the chi-squared model adjustment; df—number of degrees of freedom; *p*—calculated level of test significance; pseudo R^2^—value that evaluates explanatory variable anticipation according to the model; b_i_—coefficient estimation in the regression model; SE_i_–standard error estimation for the b_i_ coefficient; z_i_—value of test statistics in a standard distribution; (*p_i_* = Pr (>|z_i_|)—calculated probability value *p_i_* for the double-sided critical area equal to z; *n*—group quantity.

## Data Availability

The data presented in this study are available on request from the corresponding author.

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
