# Peer review of "Factors Differentiating Rural and Urban Population in Determining Anxiety and Depression in Patients with Chronic Cardiovascular Disease: A Pilot Study"

_ijerph, 2021, doi:10.3390/ijerph18063231_

Round 1

Reviewer 1 Report

The manuscript ‘Anxiety and depression in patients with chronic cardiovascular disease—factors differentiating rural and urban subpopulations’ by  Szlenk-Czyczerska et al targets identifying the environmental and social factors between rural and urban populations that impact anxiety and depression, especially in patients with cardiovascular disease history. The authors used a cross-sectional observational study design and a subjective survey, QoL assessment questionnaires, and HADS-M for depression scale for anxiety and depression assessment to address the hypothesis.  The data was analyzed using a logistical regression model to provide a cause and effect relationship between the factors and anxiety and depression. The data was mostly analyzed using non-parametric tests given the smaller sample size, larger variance, and non-normal distribution.

Overall, the authors have taken due consideration to minute details in terms of study design, implementation, data analysis, and presentation. Tables, footnotes and statistical analysis are robust and well written.

Although the study has merits in distinguishing the environmental factors variation affecting the incidence of anxiety and depression between rural and urban populations, the smaller sample size does not let it extrapolate these findings to a larger population. However, it can provide enough more robust pilot data to propose a direction to consider such factors in the future. The authors are encouraged to include the word pilot study in the title of the manuscript.

 An interesting piece of data is - not much variation in the socioeconomic status between urban and rural population as per table 3, which might be limited to the study area and might be a single largest defining factor in other geographic regions. Authors may want to discuss it in detail in discussion.

Further, the participant - gender distribution is skewed by females in both urban and rural settings. The authors might want to discuss the effects of such a sample over samples including only men vs. only women and how the gender might affect their outcome or perception of the outcome in general.

Author Response

Dear Editor,

We would like to sincerely thank the Editorial Board and the Reviewer of your esteemed International Journal of Environmental Research and Public Health for their positive feedback and constructive recommendations improving our observational paper entitled Factors differentiating rural and urban population determining anxiety and depression in patients with chronic cardiovascular disease - a pilot study by Elżbieta Szlenk-Czyczerska, Marika Guzek, Dorota Emilia Bielska, Anna Ławnik, Piotr Polański and Donata Kurpas.

W tej pierwszej rundzie weryfikacji mocno skoncentrowaliśmy nasze wysiłki na kwestiach poruszonych w Pańskim liście. Chcielibyśmy odpowiedzieć na tę opinię w oparciu o naszą dokładną weryfikację punkt po punkcie, jak widać w poniższej tabeli. W związku z tym ostateczna wersja tekstu manuskryptu zawiera wszystkie niezbędne modyfikacje i ulepszenia.

Mamy wielką nadzieję, że nasze poprawki okażą się wyczerpujące i okażą się pomocne w uzyskaniu pozytywnej ostatecznej decyzji o przyjęciu naszego artykułu do publikacji w Państwa prestiżowym International Journal of Environmental Research and Public Health .

Czekając na Twoją decyzję, pozdrawiam

Autorzy.

Reviewer 2 Report

Revision Article

Title:

Anxiety and depression in patients with chronic cardiovascular 2 disease—factors differentiating rural and urban 3 subpopulations

Thanks for reading this interesting article, with small sample.  But your manuscript can improve on:

Title and abstract

Please, indicate the study’s design with a commonly used term in the title or the abstract

Introduction

Line 39: before using an acronym it is better to describe the name in full in the definition , line 34.

Line 39: .. 24% of old people with.. Are they adult or olders people?

Line 75. Are you sure that there is no scientific research on the factors  affecting anxiety and depression comparing rural and urban CVD patient populations, in other Countries? Please, try another review research.

Line 80. Summarize the goal of the study and involve the caregivers as a contribution to the same study?

Methods

Line 93. Why the study involved caregivers? Which function for the results?

This is a cross-sectional study.  What is the sample size? How many patients for?

Line 107. Describe the methods of selection of participants (probabilistic or no sampling)?

Line 109.  If you include in the sample only adult, why in the “Introduction” describe the health problem in adolescent? this could be misleading.

Line 486. A dd …Institutional Review Board Statement: This research obtained approval from by the Bioethical 486 Commission at Medical University in Wroclaw (No KB -86/2016)…in thelast part of methods, with subtile.

Table 2° nd 3. Use the word Sex in place of Gender (there are not the same)

Table 2: Is it essential? You can describe without table in the text..use appendix for that

Results

Line 229, add fullstop after women

Line 230. … vs. is Latin language, so you can use or “versus” or “vs” both in italics, without fullstop. Add years after 76, after 101 and 94.

Line 257  You write : …anxiety and the number of visits in cardiological 257 clinics (p = 0.005)…and table 5.  Number of visits in cardiological and number of interventions of a family nurse are absent in the methods (only in table 2), please add.

Discussion

Well done

In the limits describe about comorbidities and how them impact on quality of life in this patients and in other chronic population (I advice to quote:  Paterniani A, Sperati F, et al. Quality of life and disability of chronic non-cancer pain in adults patients attending pain clinics: A prospective, multicenter, observational study. Appl Nurs Res. 2020 Dec;56:151332. doi: 10.1016/j.apnr.2020.151332 )and  other chronic population

Conclusions

Line 470: why  younger age? Pls explain better  in the results and discussion

Author Response

Chcielibyśmy serdecznie podziękować Redakcja oraz recenzentem cenionych International Journal of Badań Środowiska i Zdrowia Publicznego za ich pozytywne opinie i konstruktywnych zaleceń poprawy naszych papier obserwacyjne uprawnione Czynniki różnicujące wiejskich i miejskich populacji określającą lęku i depresji u pacjentów z przewlekłą układu krążenia choroba - badanie pilotażowe Elżbiety Szlenk-Czyczerskiej, Mariki Guzek, Doroty Emilii Bielskiej, Anny Ławnik, Piotra Polańskiego i Donaty Kurpas.

Drogi redaktorze,

W tej pierwszej rundzie weryfikacji mocno skoncentrowaliśmy nasze wysiłki na kwestiach poruszonych w Pańskim liście. Chcielibyśmy odpowiedzieć na tę opinię w oparciu o naszą dokładną weryfikację punkt po punkcie, jak widać w poniższej tabeli. W związku z tym ostateczna wersja tekstu manuskryptu zawiera wszystkie niezbędne modyfikacje i ulepszenia.

Mamy wielką nadzieję, że nasze poprawki okażą się wyczerpujące i okażą się pomocne w uzyskaniu pozytywnej ostatecznej decyzji o przyjęciu naszego artykułu do publikacji w Państwa prestiżowym International Journal of Environmental Research and Public Health .

Czekając na Twoją decyzję, pozdrawiam

Autorzy.
